# Analytical Subnetwork and IP Alias Resolution for Network Tomography Using Path Traces

Ahmet Aksoy

Department of Computer Science and Cybersecurity, University of Central Missouri, Warrensburg, MO 64093, USA; aksoy@ucmo.edu

**Abstract:** Mapping of link-level network topologies requires the processing of collected raw traces, including the resolution of alias IPs of router interfaces and underlying subnetworks. There have been several probing techniques for IP alias resolution along with an analytical resolution approach which relies on the IP address assignment practices. The analysis of IP address assignments can also reveal the underlying subnetworks, i.e., the link-level connectivity among the routers. In this paper, we present a comprehensive Analytical Subnetwork and Alias IP Resolution (ASIAR) method that relies on the analytical analysis of path traces to infer underlying subnetworks and router interfaces. While ASIAR increases the efficiency of the analytical resolution method, it integrates additional sanity checks and performs parameter adjustment based on the topology sampling characteristics to improve the resolution accuracy. We explore how different network sampling issues affect the analytical resolution, and analyze the accuracy of the ASIAR on synthetic and genuine networks. Compared with the state-of-the-art analytical resolution method, ASIAR is able to increase both precision and recall by fine-tuning the parameters of sanity checks used for analytical subnetwork and IP alias resolution.

**Keywords:** Internet measurement; network topology; traceroute datasets

## 1. Introduction

The Internet grows with an interplay between competition and coordination of network providers. As of July 2019, over 65 thousand Autonomous Systems (AS) connect individuals, businesses, universities, and agencies while focusing on optimizing their communication efficiency [1]. As each network is built by different AS for possibly different purposes, e.g., small local campus to a large transcontinental backbone provider, a single or a subset of AS(es) is not representative of the whole Internet [2]. Measuring and understanding Internet topology is essential for managing, securing, and enhancing it, as it aids network practitioners in developing new protocols and services and helps protect the national cyberinfrastructure. Researchers obtain sample network topologies by generating measurement probes and collecting a large number of path traces to observe characteristics of the underlying network [3,4]. Most topology measurement studies utilize the well-known Internet debugging tool, traceroute [5], or its variants to collect path traces from a set of vantage points [6–8]. Traceroute uses TTL-scoped probe packets to obtain ICMP error messages from the routers on the path from a local system to a given remote system. By collecting the source IP addresses from the incoming ICMP packets, traceroute returns the path as a sequence of IP addresses. Network topology is then built as a graph of router adjacency. In order to facilitate measurement studies, several research groups have developed systems to collect the required information from geographically diverse vantage points [9]. The measurement platforms include the Archipelago measurement infrastructure of CAIDA [10], RIPE Atlas [11], measurement-lab [12], and the Internet Mapping System [13].

After collecting topology data, one needs to process this data to obtain the underlying network topology [14]. Topology construction tasks include: (i) resolving unresponsive routers that are marked by asterisks ('*') in trace output, since some routers do not respond to the measurement probes [15,16], (ii) finding IP addresses belonging to the same router as routers may appear with different IP addresses in different path traces [17,18], and (iii) identifying underlying physical subnetworks among IP addresses that reveal the link-level connectivity [19]. While the term subnet is used to refer to both the connection medium and the IP address range, we use the term subnetwork to refer to the connection medium and the term subnet to refer to the specific IP address range assigned to a connection medium (i.e., a subnetwork). Inaccuracies in topology collection and construction may significantly affect the accuracy of the results obtained in the measurement study [20,21]. In general, finding the ground truth for the network topology measurement is very challenging [22]. When handling topology resolution tasks, one needs to make decisions based on the observations to infer the underlying connectivity. As the earlier decisions affect the later ones, obtaining the most likely topology under various conditions has been shown to be NP-hard [23]. Several approaches have been proposed to reduce the set of hypotheses in the decision making of the resolution tasks [24–26].

In this paper, we present the Analytic Subnetwork and IP Alias Resolution (ASIAR) approach, which performs link and router inference using the common IP assignment practices. ASIAR enhances both the efficiency and accuracy of the analytical subnetwork and IP alias resolution methods. We perform a detailed analysis of the resolution completeness and accuracy using synthetic networks as the ground truth. We analyze different topology sampling issues on the synthetic network to understand how network sampling affects the resolution results. The results indicate that differing sample networks require a different set of sanity checks and parameters to yield the best resolution performance. Hence, we introduced an evolutionary computing approach to select the utilized sanity checks and parameters on synthetic networks, which are generated based on the provided topology measurements. Finally, we analyze the performance of ASIAR on a genuine network as the ground truth.

In this study, we showed that one-fits-all approach is not suitable for analytical alias resolution as different topology samples contain varying level of detail. Prior analytical alias resolution tools utilized a single set of toolset for any network. Instead, ASIAR selects the sanity checks and optimizes the parameters based on synthetic networks of similar characteristics.

The contributions of this paper are (i) an improved and efficient analytical IP alias resolution method, (ii) an improved subnetwork inference method, (iii) a systematic analysis of factors influencing analytical subnetwork and IP alias resolution approaches, (iv) a detailed analysis of the resolution performance with respect to different sampling approaches, (v) an integrated analytical resolution approach that considers the sampled topology dataset characteristics to optimize resolution performance, and (vi) an automated parameter optimization using evolutionary computing on a synthetic network to obtain optimal parameters for the subnetwork and IP alias resolution of the given network. The ASIAR source code and experimental data are available at https://github.com/netml/asiar (accessed on 7 April 2024).

In the rest of the paper, Section 2 summarizes the related work. Section 3 provides the background on the analytical subnetwork and IP alias resolution approaches. Section 4 describes the proposed analytical subnetwork resolution approach, and Section 5 details the improved analytical IP alias resolution. Section 6 presents the automated parameter optimization of ASIAR using evolutionary computing on synthetic networks and the synthetic network generation methodology utilized for the evaluation of resolution parameters. Section 7 shows the evaluation of different analytical resolution conditions and their contribution to handling various network sampling issues. Section 8 concludes the paper.

## 2. Related Work

In this section, we present related work on IP alias and subnetwork resolution to produce link-level topologies from a sampled set of path traces.

### 2.1. IP Alias Resolution

Mercator [27] and iffinder [28] send a probe packet to a destination IP1 and if the received ICMP response is from an IP2, they assume both IPs to be aliases. However, this implementation is not common as most routers copy the IP address from the original packet. In [29], we identified only a few IP aliases using this approach, as many routers reply with the interface address targeted by the probe packet. The DNS-based method relies on the similarities in the hostnames of routers and works when an AS uses a systematic naming convention in assigning DNS names to router interfaces [30]. DNS can be utilized only when a systematic naming scheme is used by the AS and the naming template is present in the database. Another approach is to use the record route option of the IP protocol, which records up to nine IPs of routers the packet has traversed. The recorded interfaces are usually the outgoing interfaces while traceroute returns the incoming interfaces [28,31]. In earlier experiments, we showed that the record route based method is not very successful as routers generally dropped packages with the record route option set [29].

A commonly utilized approach relies on the IP identification field value in the IP protocol header of the returned ICMP error messages [3]. Typically, the IP identification field of IP header is implemented as a monotonically increasing counter [32]. Given two IP addresses, the ally tool sends successive probe packets to each of the two addresses, and a third packet to the address that responds first. If the responses have IP identification values in sequence with a small difference in between, they are likely to be aliases. Since this method requires $O(n^2)$ probes to test all possible pairs, several approaches have been deployed to reduce the number of probes [17,32,33]. Moreover, as some routers do not respond to certain probes, researchers have proposed alternative probes to increase the elicited responses from routers [34]. Pamplona-traceroute [8] integrated IPID-based alias resolution into topology discovery. RadarGun introduced an analysis of IPID velocity over time rather than pairwise probing to reduce analysis overhead [17]. More recently, the MIDAR tool has developed a parallelized version of the method by analyzing the change in velocity of IP identifiers [35]. Ref. [36] presents a method to improve IP alias resolution and dual-stack inference by using multiple protocols such as SSH and BGP, which respond to unsolicited requests with unique device identifiers after a TCP handshake. Ref. [37] introduces an alias resolution tool leveraging ICMP rate limiting for identifying router interfaces as aliases. It uses ICMP probes and machine learning to improve the accuracy of alias resolution. Ref. [38] presents a method improving router IP alias resolution by using path length comparisons from traceroutes, avoiding reliance on router-specific characteristics.

We introduced an Analytical Alias Resolution (AAR) [39] and complemented it with probing in Analytical and Probe based Alias Resolution [18]. The analytical approach relies on common IP address assignments to infer the underlying IP aliases (see Section 3.3 for details). Kapar [40] improved analysis overhead of the approach by removing conflict sets. Additionally, PalmTree [41] analyzes distance of IP addresses to vantage points to infer the underlying IP aliases. In [42], the authors used a fingerprinting process to determine the behavior of IPs that are alias candidates and determine which of the IP alias resolution techniques (i.e., TreeNet, MIDAR, and Kapar) to use for the candidate subsets.

As probing approach is orthogonal to the analytic resolution, it is important to utilize both in IP alias resolution [18]. Hence, one should use the state-of-the-art probing method, i.e., currently MIDAR, along with the introduced analytical approach of ASIAR.

### 2.2. Subnetwork Resolution

Subnetwork inference is pivotal for unveiling the underlying link-level connectivity of routers across point-to-point or multi-access links, aiding in the comprehensive mapping of internet architecture (refer to Section 3.2 for detailed insights).

On a ground truth sample of Internet2 backbone [19], we had an error rate of 7% in detecting underlying links using the analytical approach. The analytical approach does not require probing to elicit information from routers but benefits from IP address assignment practices. Additionally, it may be utilized on historical datasets where probing based IP alias resolution cannot be applied. TraceNET [6] implemented the approach as a stand-alone subnetwork tracing tool similar to traceroute and exploreNET [43] performed subnet level mapping of network topologies. TreeNET [44] further improved subnet discovery using the tree-like discovery of the networks from a remote vantage point.

### 3. Background on the Analytical Resolution Approaches

In this section, we first present the common IP address assignment practices, and then show how we utilize it to infer the underlying subnetworks [19] and IP alias resolution [18].

### 3.1. IP Address Assignment Practices

Networked devices are interconnected using a point-to-point or a multi-access link. Devices connected over the same medium form a subnetwork and share the same link-layer collision domain [45]. Devices on the same subnetwork are assigned IP addresses from the same subnet prefix. The addressing of device interfaces usually adheres to the IETF guidelines, particularly the RFC-2050 Internet Registry IP Allocation Guidelines. An AS obtains a range of IP addresses with the same subnet prefix, which is divided into smaller chunks to be assigned to individual interfaces connected to the same subnetwork. The hierarchical assignment helps in reducing the routing table sizes. We utilize this IP address assignment practice to infer the subnetworks and resolve IP aliases.

The smallest subnetwork is a point-to-point link with two device interfaces and needs a /30 or a /31 subnet (the latter is introduced in RFC 3021). Larger subnet ranges (e.g., /29 or bigger) would waste IP addresses, a scarce commodity at IPv4, and hence, they are usually avoided for point-to-point links. On the other hand, multi-access links connect more than two devices and require a subnet range that is /29 or larger. For instance, in a /29 subnet address, we have $2^{(32-29)} - 2 = 6$ IPv4 addresses for assignment, where the first one is used as the network address and the last one is used as the subnetwork broadcast address.

In general, a subnetwork with $k$ devices needs a $/x$ subnet address, where $x = 32 - \lceil log_2(k+2) \rceil$. The first $x$ bits of the IP addresses denote the subnet address, and the last $32 - x$ bits identify the interface connected to the subnetwork. For instance, 192.168.0.0/28 subnet has the last four bits to identify the individual IP addresses of the device interfaces. These four bits can identify at most 14 interfaces connected to a subnetwork. The remaining two IP addresses, namely 192.168.0.0 and 192.168.0.15, are the subnetwork and the broadcast addresses, respectively, and typically are not assigned to interfaces.

### 3.2. Subnetwork Inference

Subnetwork resolution identifies the underlying connectivity among router interfaces to build link-level topologies from collected path traces [19]. Routers are connected over point-to-point or multi-access links, and the subnetwork resolution helps in identifying the underlying link structure. In the subnetwork resolution task, the IP addresses in a dataset are analyzed to infer the subnetwork relations among them. This inference also helps to detect the link-level connectivity between IP addresses, denoted with ↔ in the rest of the paper.

As an example, consider four routers *A*, *B*, *C*, and *D*, shown in Figure 1a, that are connected to each other via a multi-access link. Assume that the collected set of path traces

includes the *A–B* link and the *B–C* link and no path trace at hand include the *A–C* link or any link between the router *D* and others from the subnetwork. In this case, a link-level map that does not consider the subnetwork relation among these IP addresses will yield a subgraph, as shown in Figure 1b, which is considerably different from the underlying topology. A careful study of the IP addresses may detect the subnetwork relation between the router IPs and improve the resulting map with the inclusion of the missing links, as shown in Figure 1c.

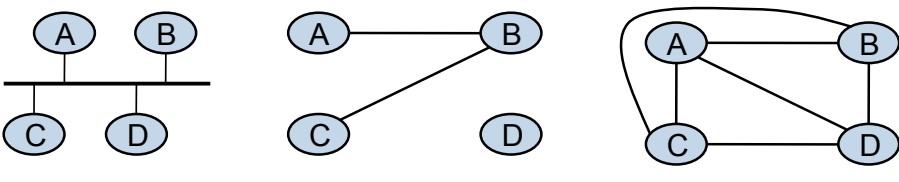

(a) Multi-access link   (b) Observed topology   (c) Point-to-point model

**Figure 1.** Subnetwork resolution.

The goal in subnetwork resolution is to identify multiple links that appear to be separate and combine them to reveal their corresponding single-hop connection medium (i.e., point-to-point or multi-access link) [19]. Subnetwork resolution also finds the missing links between IP addresses that fall in the same subnetwork range but were not observed in the path traces. For instance, in [29], subnetwork identification helped with the addition of 16% more links that were not directly found in the path traces. Hence, the successful inclusion of subnetwork relations among the routers yields topology maps that are closer to the sampled segments of the networks at the link layer.

In order to identify link-level topology map from path traces, we need to analyze the measurement dataset to infer subnet relations among the IP addresses. The IP address assignment practices summarized in the previous subsection led to a subnet relation among IP addresses connected to the same physical subnetwork. We can infer subnetworks where IP addresses can be grouped into a subnet address range under an address prefix of length $/x$. On the other hand, any two IP addresses can be grouped into an address range for a sufficiently large subnet range (i.e., an amply small $/x$ prefix length). Hence, we need to look for evidence in measurements that two IP addresses are not in the same physical subnetwork.

Each observed IP address belongs to some subnet where all interfaces on the subnet have IP addresses with the same maximal $x$ bit prefix. IP addresses of the subnet interfaces have the same $x$ bit prefix and no other interface IP address has the same $x$ bit prefix. Note that a loopback IP address has a subnet of /32, as it needs only one IP address to identify the device. Based on this observation, we introduced an iterative approach to identify subnetworks in [19]. We first form all candidate $/x$ subnets from the dataset by combining the IP addresses whose first $x$ bits match. Next, we recursively construct smaller subnets (e.g., $/x$, $/(x+1)$, ..., /31 subnets) while there is lack of evidence for a physical subnetwork among the subnet IP addresses. We need to detect and prune candidate subnets that do not correspond to physical subnetworks, and, in [19], we developed a set of complementary conditions to verify subnetworks, shown below.

Condition 1—Subnetwork Accuracy: Given a loop-free path trace, two or more IP addresses from the same subnetwork cannot appear in any path trace without having a successor/predecessor relationship with each other. That is, IP addresses in a subnetwork (or their aliases) should appear consecutively if they appear in the same trace. For instance, consider the sample topology in Figure 2, where $a \cdots d$ are end-hosts and $A \cdots E$ are routers. Without the knowledge of network topology, a path trace from *a* to *b*, i.e., (2, 10, 4), will indicate that interface IPs 2 and 4 cannot be in the same subnetwork. as they are two hops away of each other in a trace. While RFC 1812 states that ICMP error messages should be sent with the IP address of the outgoing interface toward the vantage point, another interface's IP might be returned. This practice could yield path traces with two

IP addresses of the same subnetwork appear consequently, such as interfaces 9 and 10, if router A was to reply with IP address of 9 instead of the expected IP of 2. Hence, the accuracy condition allows a subnetwork with IP addresses that appear consecutively in a trace but not farther apart.

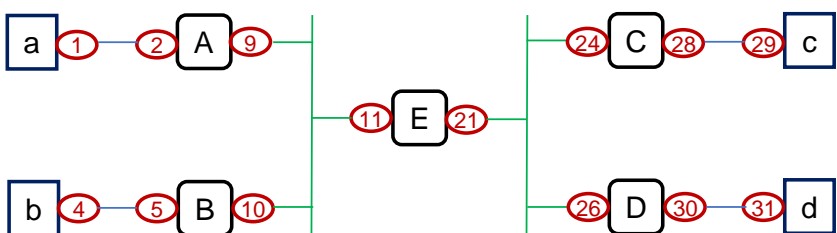

**Figure 2.** Sample network with point-to-point and multi-access links connecting.

Condition 2—Subnetwork Distance: Given a candidate /$x$ subnetwork, IP addresses of the subnetwork should be at a similar distance to vantage points. For instance, for the same trace between *a* to *b* in Figure 2, we would observe that interface IP 2 is at one hop to host *a* but interface IP 4 is three hops away.

Condition 3—Completeness: Completeness is a measure that determines how many of the subnet IPs are observed in the collected traces. The condition ignores candidate subnetworks that have less than a given fraction (e.g., one quarter or half) of their IP addresses present in the collected dataset. A /$x$ subnet can include up to $2^{32-x} - 2$ IP addresses and observing a sparse set of IP addresses in a subnet may lack evidence to verify the conditions 1 or 2. Without this requirement, it would be easy to form a large subnetwork using a few IP addresses. However, with only a few observances, it is difficult to verify the existence of a physical subnetwork. Depending on the completeness ratio, this condition may cause us to discard a real subnetwork of size, say /25, and instead, to consider one or more smaller subnets that satisfy the completeness criteria.

After assessing the previous conditions, we ignore candidate subnetworks that are a subset of bigger candidate subnetworks. IP addresses of a /$x$ subnet could appear in a smaller (e.g., /($x + 1$), /($x + 2$), etc.) subnet, and we accept the largest one as valid. The only exception is when an IP address appears to be in both a /30 and a /31 candidate subnets. In this case, /31 subnetwork is chosen as the valid one.

### 3.3. IP Alias Identification

As routers have multiple interfaces, each interface typically has a unique IP address. Note that an interface could borrow another interface's IP address through an IP unnumbered mechanism [46]. Additionally, a globally unique loopback address might be assigned to the router using a /32 subnet [47]. In a given set of path traces, a router may appear on numerous path traces with different IP addresses. Therefore, there is a need to identify and group IP addresses belonging to the same router, denoted with ≡. Without the IP alias resolution, the resulting network graph may be significantly different from the underlying topology [20]. For instance, in Figure 3a, each router has multiple interfaces with unique IP addresses. Collecting traces between all pairs of *X*, *Y*, and *Z* end systems, we would obtain a sampled topology as in Figure 3b without the alias IP resolution. We need to identify IP aliases for each router and cluster them as shown with the green circles. Several studies pointed out the impact of incomplete IP alias resolution in certain measurement studies [20,48]. Varying the resolution success rate, we showed that the IP alias resolution process has a significant impact on the observed topological characteristics [20].

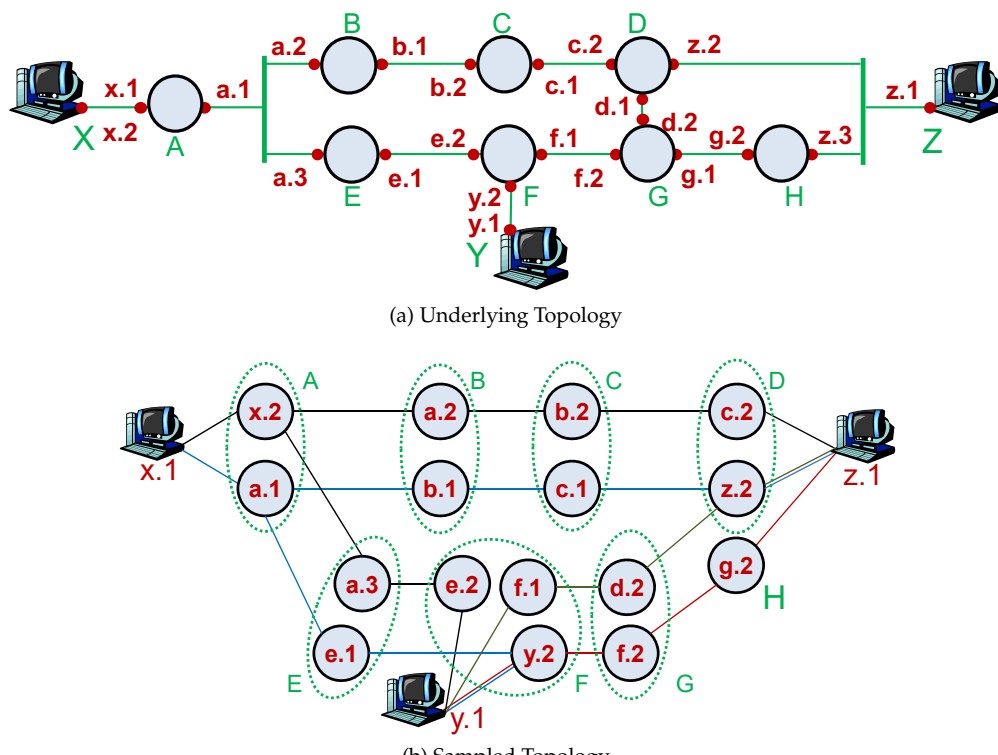

(a) Underlying Topology

(b) Sampled Topology

**Figure 3.** IP aliases in a sampled topology.

We introduced an analytical IP alias resolution methodology based on the IP address assignment practices [18,39]. After inferring the underlying subnetworks, we can identify IP aliases when multiple traces cross the same subnetwork from different directions. For instance, given a trace from $a$ to $b$ and $b$ to $a$ in Figure 2, we would have traces of (2, 10, 4) and (5, 9, 1). Reverse alignment of the traces would lead to:

```
(2, 10, 4)      trace from a to b
(1, 9,  5)      reversed trace from b to a
```

This alignment, dictated by the subnetwork relation among 2↔1, 10↔9, and 4↔5, not only reveals alias pairs of 2≡9 for router *A* and 10≡5 for router *B*, but also allows us to infer IP aliases from path traces rooted in the subnetwork relation between IP addresses of trace segments. Moreover, while the alignment of reversed trace routes is possible based on these subnetwork relations, there exist specific conditions that help negate any misalignments. This method significantly improves our understanding of network topology by accurately identifying routers and their connections within the vast expanse of the Internet, thereby enhancing network management and security measures.

As there would be conflicting observations, the processing order of aliases would influence the results. Hence, during IP alias resolution, we process IP aliases introduced by subnetworks with a higher completeness ratio, as they have a stronger verification compared to a subnetwork with lower completeness. For subnetworks with the same completeness, priority is given to the ones involving more path traces. Note that, by definition, all /31 and /30 subnets are 100% complete and are processed before other subnets.

Condition 4—IP Alias Accuracy: Assuming that path traces are loop-free to start with, the inferred IP alias pairs should not introduce any routing loop in any of the path traces. Additionally, no pair of subnetwork IP addresses should be set as aliases. While there could be multiple interfaces of a device connected to the same subnetwork to increase its throughput, this is an extremely rare practice. Note that in a recent study [49], we observed that 4.4% of path traces have a routing loop within an autonomous system, but we assume path traces to be loop-free when IP aliases are introduced.

Condition 5—IP Alias Distance: Given two IP addresses that are candidate aliases for a router, they should be at the same distance to the vantage points. The distance check helps to improve the accuracy of the resolution. Different from subnetwork distance, in the IP alias resolution, we require IPs to be at the same distance.

Condition 6–Common Neighbor: Given two IP addresses that are candidate aliases (e.g., 2 and 9 in Figure 2) for a router (i.e., *A*) because of subnetwork relation on one side of the alignment, one of the following rules should hold for the other side of the alignment in order to set the IP pair as an alias:

(i)     They have a common neighbor in any path trace;
(ii)    There exists a previously inferred IP alias pair (e.g., 5≡10) such that 5 is a successor (or a predecessor) of 2 and 10 is a predecessor (or a successor, respectively) of 9;
(ii)    The involved path traces are aligned such that they form two subnetworks, one at each side of the router.

This rule helps to avoid misalignment of a subnetwork of a multi-access link as multiple pairs of subnetwork IPs could be aligned. For instance, a trace of (2, 10, 4) from *a* to *b* could be aligned with a trace of (28, 21, 9, 1) from *c* to *a* as follows:

```
    (2, 10,  4)    trace from a to b
(1, 9, 21, 28)    reversed trace from c to a
```

In this alignment, we have two subnetworks 2↔1 and 10↔9. Since alias pair 2≡9 is between two subnetworks, we could set it correctly. However, 10blue≡21 pair has only one neighboring subnetwork between 10blue↔9, while the other side lacks an alias (i.e., between 4blue≡28) or a subnetwork (i.e., between 4blue↔21). Hence, this condition would avoid incorrectly setting 10blue≡21 as an alias pair.

## 4. Analytic Subnetwork Resolution

In this study, we extend Ref. [19] by revising the conditions that are utilized for subnetwork resolution. The distance condition that required IP addresses of a subnetwork to be at similar distances was employed with respect to vantage points. In this study, we extend the distance condition to be with respect to the reference points, which include ingresses and egresses of the AS in addition to any vantage point that might be in the AS. This allows for a better granularity in tabulating IP distances as it eliminates trace segments through the neighboring ASes. Given that AS-level paths are much more dynamic than router level paths within an AS [49], this change reduces the impact of path dynamics on the distances.

Resolving IP aliases of ingresses and egresses would help in reducing the number of columns in the distance matrix and more importantly merge the information of IPs of the same ingress or egress router. Hence, to improve resolution accuracy and computation efficiency, one should utilize the IP identifier-based probing technique to resolve alias IPs before running the analytical subnetwork resolution method. We also update the IP completeness condition, which requires a certain percentage of subnetwork IPs to be observed, to use thresholds rather than a fixed value.

Line 1 parses traces into IPs array of length *numIP* and Distances of length *NumRP* by *NumIP* data structures. ParseIPs obtains trace segments of the given AS# based on the BGP announcements, which include subnetwork regions from the AS. If the trace did not start from a vantage point within the AS, the first IP of the trace segment is considered as an ingress point. Algorithm 1 presents the pseudocode for the identification of subnetworks.

---

**Algorithm 1** SubnetworkIdentification(*Traces, BGP, AS#*)

---

1: (*IPs*, *Dists*) ← **ParseIPs**(*Traces, BGP, AS#*)
2: *IPs[0].bound* ← true
3: *IPs[numIP].bound* ← true
4: **for** *i* ← 2 to numIP **do**
5:     *IPs[i-1].bound* ← false
6:     **if** *Mask*(*IPs[i-1], IPs[i]*) < 31 **then**
7:         **for** *j* ← 0 to numRP **do**
8:             **if** |*Dists[j][i] − Dists[j][i-1]*| > 1 **then**
9:                 *IPs[i-1].bound* ← true
10:                 **break**
11:             **end if**
12:         **end for**
13:     **end if**
14: **end for**
15: *GroupDistanceCheck*(*IPs, Dists*)
16: *CompletenessCheck*(*IPs, Dists*)
17: *Subnets* ← *SortedSubnets*(*IPs*)

---

Similarly, the last non-AS IP, if any, is assumed to be an egress point. For example, in a trace of (A, B, C, D, E, F, G) where IP addresses *C*, *D*, and *E* belong to the AS, IP *B* is considered an ingress of the AS, while IP *F* is marked as egress. The distances of all IPs within the trace segment is updated with respect to the Reference Points, which are (i) vantage or ingress point and (ii) egress point if any.

The algorithm then considers every consecutive pair of IPs and checks whether they should be in separate subnetworks based on the distances lines 3–8). As a subnet mask of /31 indicates a point-to-point link, IPs within a /31 subnet are not separated (line 5). If for any reference point, the distance between IPs is more than one hop, then the IPs are separated (lines 7–8). After separating consecutive IPs that could not be in the same subnetwork, the algorithm checks whether the IPs within each subnetwork group satisfies the distance condition using the GroupDistanceCheck, presented in Algorithm 2. After identifying potential subnetwork boundaries, it checks if the blocks of IPs satisfy the completeness condition, presented in Algorithm 3. Finally, subnets are ranked by their completeness and observation frequency (line 11).

Algorithm 2 checks if IPs within a candidate subnetwork satisfy the distance condition as a group or whether they should be separated into multiple subnetworks. It first identifies the current group's boundaries (lines 1–3) and then analyzes the IP block with respect to each reference point (lines 5–6). If the distances between any two IPs are greater than one hop for any reference point, the IPs are separated into different subnetworks (line 10). The Mask function in line 11 identifies the maximal subnet mask that would separate the conflicting IPs. The loop in lines 13–15 determines the boundary between both IPs. The loop in lines 17–19 identifies the boundary between the smaller IP and the upper region of the IP list so that a larger subnet does not contain both IPs. Similarly, the loop in lines 21–23 identifies the boundary between the larger IP and the lower region. If there is any division while checking the current IP block, the outer loop backtracks with a new end position (line 25). If there is no division within the current IP block, then the outer loop advances to check the next IP block for the distance condition (line 26).

---

**Algorithm 2** *GroupDistanceCheck(IPs, Dists)*

---

1: *start* ← 0
2: *end* ← 0
3: **while** *start* < *numIP* **do**
4:     **while** !IPs[++*end*].bound **do**
5:         *goBack* ← false
6:         **for** *j* ← [0, RP) **do**
7:             **for** *i* ← [*start*, *end*] **do**
8:                 *min* ← ∞; *max* ← 0
9:                 **if** *Dists[j][i]* < *min* **then**
10:                     *min* ← *Dists[j][i]*; *minP* ← *j*
11:                 **end if**
12:                 **if** *Dists[j][i]* > *max* **then**
13:                     *max* ← *Dists[j][i]*; *maxP* ← *j*
14:                 **end if**
15:                 **if** *max* - *min* > *1* **then**
16:                     *match* ← **Mask**(*IPs[minP]*, *IPs[maxP]*)
17:                     *from* ← max(*minP*, *maxP*); *to* ← *from*
18:                     **while** + + *to* ≤ *end* **do**
19:                         **if** **Mask**(*IPs[from]*, *IPs[to]*) ≤ *match* **then**
20:                           *IPs[to − 1]*.bound ← true; **break**
21:                       **end if**
22:                   **end while**
23:                   *to* ← *from*
24:                   **while** − − *to* ≥ *minP* **do**
25:                     **if** **Mask**(*IPs[from]*, *IPs[to]*) ≤ *match* **then**
26:                         *IPs[to]*.bound ← true; **break**
27:                     **end if**
28:                   **end while**
29:                   *from* ← *to*
30:                   **while** − − *to* ≥ *start* **do**
31:                     **if** **Mask**(*IPs[from]*, *IPs[to]*) ≤ *match* **then**
32:                         *IPs[to]*.bound ← true; **break**
33:                     **end if**
34:                   **end while**
35:                   *goBack* ← true
36:                   **break**
37:                 **end if**
38:             **end for**
39:             **if** *goBack* **then break**
40:             **end if**
41:         **end for**
42:     **end while**
43:     **if** !*goBack* **then** *start* ← *end* + 1; *end* ← *start*
44:     **end if**
45: **end while**

---

Algorithm 3 checks for the completeness condition of subnetworks to assure that they have a sufficient number of IPs observed in the trace dataset. For each subnetwork that is not point-to-point (i.e., /30 or /31), the algorithm counts the number of observed IPs from the IP block with compl (line 7) and the number of observed IPs for each reference point with refCompl (lines 8–13) variables. If either is below the selected thresholds, it divides the IP block into two subnets (lines 14–17). The observation threshold checks the number of IP observations with respect to a particular reference point.

---

**Algorithm 3** *CompletenessCheck(IPs, Dists)*

---

1: *start* $\leftarrow$ 0
2: *end* $\leftarrow$ 0
3: **while** *start* < *numIP* **do**
4:     **while** !IPs[++*end*].bound **do**
5:         *goBack* $\leftarrow$ false
6:         *mask* $\leftarrow$ **Mask**(*IPs*[*start*], *IPs*[*end*])
7:         **if** *mask* < 30 **then**
8:             *matches* $\leftarrow$ 0; *compl* $\leftarrow$ (*end* $-$ *start*)$/2^{(32-mask)} - 2$
9:             **for** $i \leftarrow [0, \text{RP})$ **do**
10:                 *obs* $\leftarrow$ 0
11:                 **for** $j \leftarrow [start, end]$ **do**
12:                     **if** Dists[j][i] $\neq \varnothing$ **then** *obs* $++$
13:                     **end if**
14:                 **end for**
15:                 **if** *obs*$/(end - start) >$ obsThld **then** *matches* $++$
16:                 **end if**
17:             **end for**
18:             *refCompl* $\leftarrow$ *matches*$/(31 - mask)$
19:             **if** *compl* < compThld | *refCompl* < refThld **then**
20:                 **for** $i \leftarrow (start, end]$ **do**
21:                     **if** **Mask**(*IPs*[*start*], *IPs*[*i*]) $\leq$ *mask* **then**
22:                         *IPs*[*i* $-$ 1].bound $\leftarrow$ true;
23:                         *goBack* $\leftarrow$ true
24:                         **break**
25:                   **end if**
26:                 **end for**
27:             **end if**
28:         **end if**
29:         **if** !*goBack* **then** *start* $\leftarrow$ *end* $+ 1$; *end* $\leftarrow$ *start*
30:         **end if**
31:     **end while**
32: **end while**

---

## 5. Analytic IP Alias Resolution

Analytic and Probe-based Alias Resolver (APAR) [18] utilizes the common IP address assignment scheme (RFC 2050) to infer IP aliases. Given a set of path traces, APAR uses inferred subnets to align symmetric segments of different path traces and identifies alias pairs among involved IP addresses. Path asymmetry is a commonly observed characteristic on the Internet. However, the approach does not require complete path symmetry and relies on symmetric path segments to resolve IP aliases. Additionally, APAR uses a lightweight probing component, i.e., $O(n)$ probes, to improve its accuracy. APAR is a scalable approach that could work with historical datasets and produced IP aliases that had a false positive rate of less than 10% [18].

As pointed out in the Kapar enhancement [32], original APAR implementation had high storage requirements. In this study, we modified the neighbor matching to focus on neighboring subnets, as shown in Figure 4. The red arrows indicate a subnetwork relation between IP addresses from two different path segments of (IP1, IP2) and (IPb, IPa), and the green circle indicates the alias IPs. In the proposed modification, we identify IP aliases that are between two subnets without requiring to record the triplets as in [32]. We also included additional sanity checks as indicated below, where a subset of the modes is selected based on the sampled network topology characteristics.

These changes facilitate the elimination of the necessity to retain complete path traces in memory for IP alias resolution, significantly enhancing the processing of candidate aliases. When evaluating an alias pair, as illustrated in Figure 4, there exists an alternative

alignment where IPa is an alias to IP2, necessitating extra conditions to ascertain the most plausible alignment for accurately identifying the correct alias pair. This strategic refinement in methodology not only streamlines the data management process but also elevates the precision of network mapping by discerning the true connections between IP addresses, thereby refining our approach to network analysis and security.

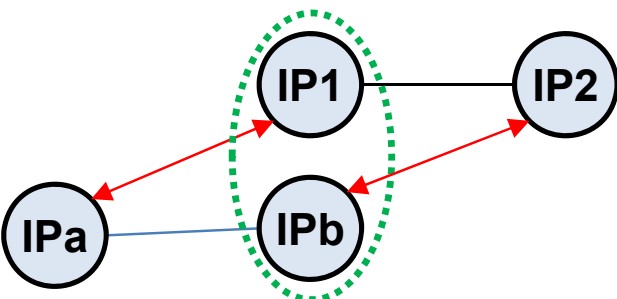

**Figure 4.** IP aliases.

Algorithm 4 identifies IP aliases from identified subnetworks, starting with *ParseEdges* to collect AS edges and path trace sources and destinations (line 1). This initial step forms the groundwork for a deeper analysis into the network's structure, enabling the algorithm to efficiently uncover IP aliases by understanding how various network segments interconnect. Edges contain the set of edges where $e = (IP_1, IP_2)$ and both $IP_1$ and $IP_2$ belong to the BGP announcements of the AS. *SortEdges* sorts the edges by the geometric mean of the completeness and the reference completeness of the subnetworks of the IPs on both sides of the edge (line 2). Depending on the number of sources and destination IPs of the network, the modes which are to be considered when performing IP alias resolution is determined (line 3). The modes depending on the network characteristics as explored in Section 7. For each mode to be considered (line 4), the edges are iterated one by one to determine IP aliases (line 5). For each edge $e_1$ considered, the algorithm finds the edges $e_1^{pairs}$ to be compared to $e_1$ (line 6). The edges in $e_1^{pairs}$ have at least one IP which is in the same subnetwork as one of the IPs of $e_1$.

*SortEdges* then sorts the edges in $e_1^{pairs}$ by the geometric mean of the completeness and the reference completeness of the subnetworks of the IPs on both sides of the edge (line 7). For each edge in $e_1^{pairs}$ (line 8), the algorithm initially checks which of its IPs are in the same subnetwork as $IP_1$ of $e_1$. The IP that is in the same subnetwork as $IP_1$ is named $IP_a$, and the other one is named $IP_b$ (lines 9–12). Next, potential IP aliases are formed as $pair_1$ and $pair_2$ (line 13). Note that, depending on the alignment of subnets in Figure 4, we can have a pair of aliases as $IP_1 \equiv IP_b$ or $IP_2 \equiv IP_a$.

The algorithm then runs the DistanceMatch for $pair_1$ and $pair_2$ in line 14. Distance-Match function checks and returns the number of reference points that observed the IPs of the pair at the same distance. If any of the reference points reach the IPs at different hops, the function returns *false*. If none of the reference points observe both of the IPs, the function returns null. Note that if either of the IPs is already aliased with another IP, the combined distances are utilized for distance check. In line 15, the algorithm runs the NonConflictMatch for $pair_1$ and $pair_2$. NonConflictMatch function checks whether or not the IPs in the pair were at conflict. Then, the algorithm calls the CheckAlias function along with the modes to check whether either of the potential alias pairs satisfies the conditions and is to be added to the Aliases list.

---

**Algorithm 4** AliasResolution(*Traces*, *IPs*, *Dists*, *Subnets*)

---

1: (*Edges*, *Sources*, *Destinations*) ← **ParseEdges**(*Traces*)
2: *SortEdges*(*Edges*, *Subnets*)
3: *Modes* ← **findModes**(*IPs*, *Sources*, *Destinations*)
4: **for** ∀mode ∈ Modes **do**
5:    **for** ∀$e_1$ ∈ Edges **do**
6:       $e_1^{\text{pairs}}$ ← **GetSubnetEdges**(*Edges*, $e_1^{IP_1}$, $e_1^{IP_2}$)
7:       *SortEdges*($e_1^{pairs}$, *Subnets*)
8:       **for** ∀$e_2$ ∈ $e_1^{\text{pairs}}$ **do**
9:          **if** **inSub**($e_1^{IP_1}$, $e_2^{IP_1}$)|**inSub**($e_1^{IP_2}$, $e_2^{IP_2}$) **then**
10:             $IP_a$ ← $e_2^{IP_1}$
11:             $IP_b$ ← $e_2^{IP_2}$
12:          **else**
13:             $IP_a$ ← $e_2^{IP_2}$
14:             $IP_b$ ← $e_2^{IP_1}$
15:          **end if**
16:          $pair_1$ ← ($e_1^{IP_1}$, $IP_b$)
17:          $pair_2$ ← ($e_1^{IP_2}$, $IP_a$)
18:          ($pair_1^{\text{dist}}$, $pair_2^{\text{dist}}$) ← **DistanceMatch**($pair_1$, $pair_2$)
19:          ($pair_1^{\text{noConf}}$, $pair_2^{\text{noConf}}$) ← **NonConflictMatch**($pair_1$, $pair_2$)
20:          **if** **CheckAlias**(*Aliases*, mode, $pair_1$, $pair_2$) **then**
21:             *Aliases* ← *Aliases* ∪ $pair_1$
22:          **else if** **CheckAlias**(*Aliases*, mode, $pair_2$, $pair_1$) **then**
23:             *Aliases* ← *Aliases* ∪ $pair_2$
24:          **end if**
25:       **end for**
26:    **end for**
27: **end for**

---

Table 1 outlines modes for validating candidate IP alias pairs using conditions such as IP alias sanity checks, comparing potential pairs on distance and accuracy. Unconsidered conditions are marked with a '-'. The table specifies necessary conditions for each mode, affecting performance as analyzed in Section 7. Other combinations, not offering advantages in simulations, were excluded from the discussion.

**Table 1.** IP alias resolution modes using different condition combinations.

| Mode | Common Neighbor | Distance | | Accuracy | |
|:---:|:---:|:---:|:---:|:---:|:---:|
| | | **Pair**$_a$ | **Pair**$_b$ | **Pair**$_a$ | **Pair**$_b$ |
| 1 | Subnet | ✓ | × | ✓ | × |
| 2 | Subnet | ✓ | × or ? | ✓ | × |
| 3 | Subnet | ✓ | × | - | - |
| 4 | Subnet | ✓ | × or ? | - | - |
| 5 | Subnet | - | - | ✓ | × |
| 6 | Alias | ✓ | × | ✓ | × |
| 7 | Alias | ✓ | × or ? | ✓ | × |
| 8 | Alias | ✓ | × | - | - |
| 9 | Alias | ✓ | × or ? | - | - |
| 10 | Alias | - | - | ✓ | × |
| 11 | - | ✓ | - | - | - |
| 12 | - | - | - | ✓ | - |
| 13 | - | ✓ | - | ✓ | - |

Recall that when considering an alias IP pair due to a subnet relation, there are two alignments, resulting in two possible IP alias pairs. Hence, we need sanity checks (i.e., common neighbor, distance, and accuracy conditions) to determine which is more likely. The common neighbor condition indicates whether a subnet or an alias is needed for the other side of the edge. The column indicates whether the pair of IP addresses considered as an alias is wrapped within another subnet or an alias. In case of point-to-point links, the common neighbor requirement is ignored, as there is only one pair of IP addresses within a subnet to be considered for the alignment and there is only one alignment of the subnet IPs in any trace they are involved with. Modes 1–5 require the IPs both sides of the pair to be in a subnet (as in Figure 4); modes 6–10 require the IPs on the other side to have an alias as a neighbor (e.g., an alias of IP2 is observed while IP2 is not necessarily in the same subnetwork as IPb in Figure 4); and modes 11–13 ignore the neighbor condition for point-to-point links.

The distance condition checks the potential aliases of both IP pairs, i.e., possible alignments due to a subnetwork, for a distance match. In contrast to previous research, this study employs ingress and egress points as references, based on a detailed analysis that demonstrated their greater reliability compared to a vantage point. In particular, in a recent study [49], we showed that intra-AS paths are highly dynamic while inter-AS paths are more consistent. There are three possibilities with the distances of candidate alias IPs; (i) If potential IP aliases are observed at different distances by one or more reference points, the distance condition is not met and is marked as '×', (ii) If any of the reference points do not observe both of the IPs at the same time, the distance is unknown and marked as '?'; (iii) if the distance condition is met, indicated with a '✓', where the distance of both IP addresses are the same for at least one reference point and there is no reference point that yields conflicting distances.

Finally, the accuracy condition checks if aliased IPs appear in the same trace considering the whole trace dataset. Hence, either there is a conflict due to IPs appearing in any of the traces, or they do not appear together in any of the traces.

In this paper, different from earlier studies, we implemented different combinations of sanity checks and developed an evolutionary algorithm that perform a subset of them based on the characteristics of the given network topology. In particular, we realized that different sample topologies required a different set of checks to optimize both precision and recall. Hence, rather than performing all sanity checks simultaneously, using evolutionary computing, we explore combinations of the conditions and different ordering of the modes to increase IP alias resolution performance, as detailed in the next section.

## 6. Network-Specific Parameter Optimization

Internet topology measurements obtain topologies with various density of path traces, and hence, a single parameter is not suitable in all cases. In order to assure the most optimal parameters to be used with every network that ASIAR is run on, we use evolutionary computing to automatically detect the best parameters on a synthetic network that mimics the input network generated by SONET [50,51].

We first determine the number of sources, destinations, and IPs within the input network. Then, we generate a network with the same network size to simulate traces on the input network using a similar number of sources and destinations. We run ASIAR on the sampled topology from the synthetic network with different parameter sets in order to achieve the best set of parameters for subnetwork and IP alias resolution. The synthetic network generation process is explained in detail in Section 6.1. To avoid having to run ASIAR for every possible combination of values for the parameters, we utilize a genetic algorithm (GA) to automate parameter search and to make the process as efficient as possible. The parameter optimization using evolutionary computing, i.e., GA, is explained in detail in Section 6.2.

### 6.1. Simulation on a Synthetic Network

We obtain synthetic networks that contain subnetworks and routers with multiple interfaces using the Subnetwork Oriented NETwork (SONET) topology generator [50,51]. The IP distribution for the 1000 and 10,000 node networks are provided in Figure 5 using a Hilbert curve plot. The yellow marks are the IPs assigned, the red marks are the IP space left empty while increasing the mask of the subnet needed, and the black marks are the unassigned IPs.

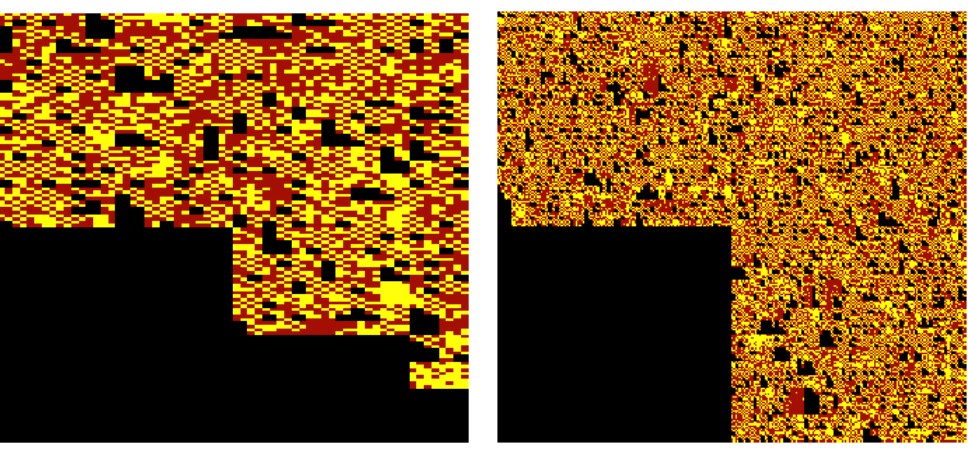

(**a**) 1000 node network.              (**b**) 10,000 node network

**Figure 5.** IP space allocation of the synthetic networks.

We then collected path traces using a number of random source and destination nodes in the generated synthetic network. The number of sources and destinations is obtained from the genuine topology data to mimic the sampling density. During path tracing, we can utilize unit edge weights to obtain the shortest paths or weighted edges to obtain the non-shortest paths. In some cases, ASes tend to route packets over longer paths due to traffic engineering rather than forwarding packets over the optimal shortest paths [49]. Hence, we obtain similar non-shortest paths by assigning power-law based edge weights. As observed path traces are not the shortest paths even within the network of an Autonomous System [49], we implemented a power-law edge weight distribution to obtain non-shortest paths while tracing synthetic networks. For instance, 10% of the generated edge weights were set to two units, 1% was set to three units, 0.1% was set to four units, and so on. Such a weighted graph produced non-shortest paths during path tracing, which replicated observation of the path characteristics observed in [49].

Additionally, as not all routers are responsive, such unresponsive routers are marked as an asterisk ('*') in the traceroute output [15]. Hence, we consider when a subset of routers is set as unresponsive routers with a power-law length distribution based on the measurements from [52]. In particular, we generated path traces when approximately 10% had one, 1% had two consecutive, 0.1% had three consecutive, and so on unresponsive routers in the path.

### 6.2. Automated Parameter Optimization Using Evolutionary Computing

A Genetic Algorithm (GA) is a computational model based on the process of natural selection inspired by evolution. A chromosome-like data structure is adapted to represent a potential solution and operations to the bits contained in the chromosomes are performed while preserving information-gaining knowledge encoded in the chromosome.

There exist three parameters that need to be optimized for subnetwork resolution. For IP alias resolution, we need to determine the modes to be considered among possible combinations of conditions and the order in which they need to be executed. Note that 59 out of 72 modes that did not contribute to any of the simulations were ignored. For the chromosome length in GA, we dedicate four bits for the representation of each subnetwork

parameters and IP alias resolution modes. Since there are 3 subnetwork parameters and 13 alias resolution modes, the chromosome length is $16 \times 4 = 64$. Since the consideration of $2^{64}$ combinations is not practical, we utilize a GA to efficiently detect as optimal values for the parameters as possible.

To determine the contribution of a potential parameter set to the subnetwork and IP alias resolution, we implemented a fitness function that GA uses to test the contribution of a chromosome. We observed cases where the F-measure of two possible solutions are very similar, yet the precision and recall differ considerably. Since we are concerned with as high precision as possible and f-measure weights the precision and recall at the same level, we preferably implemented the weighted fitness function. The fitness function used with ASIAR is formulated as: $Fitness = w_1 \times Precision + w_2 \times Recall$. To prioritize precision over recall in our subnetwork and IP alias resolution efforts, the weights are set at $w_1 = 0.8, w_2 = 0.2$. However, the user can determine the values for these parameters depending on how much prioritization for the precision and recall is desired.

GA determines optimal solution parameters for the subnetwork and IP alias resolution, while ASIAR is run on the network sampled from the synthetic network. After determining the optimal parameters on the sampled synthetic network using evolutionary computing, ASIAR is executed on the actual sampled topology data to obtain high subnetwork and IP alias resolution results, as presented in the next section.

## 7. Performance Evaluation

In this section, we present the simulation results with various synthetic networks considering different sampling approaches. We also compare ASIAR with the state-of-the-art analytical IP alias resolution tool Kapar [40]. Futhermore, we present the results on a genuine network and compare ASIAR with the state-of-the-art analytical IP alias resolution tool Kapar [40].

### 7.1. Synthetic Network Generation

We utilized the SONET topology generator to obtain synthetic networks with power-law distributions for the subnetworks and IP aliases [50,51]. We generated five networks, each containing 1000 routers, which included around 1900 IP addresses and 600 subnets. Note that while we can specify the number of routers in SONET, it determines the subnetwork and IP address distributions based on public Internet measurements and generated 1886 to 1899 IP addresses in the synthetic networks. Then, we randomly selected source and destination IPs and performed path tracing from each of these source IPs to each of the destination IPs. We used different source–destination combinations, i.e., $1 \times 1000$, $1 \times$ all, $10 \times 1000$, $10 \times$ all, $100 \times 1000$, and $100 \times$ all of the path traces as sample data.

During path tracing, we can utilize unit edge weights to obtain the shortest paths or weighted edges to obtain the non-shortest paths. In some cases, ASes tend to route packets over longer paths due to traffic engineering rather than forwarding packets over the optimal shortest paths [49]. Hence, we obtain similar non-shortest paths by assigning power-law-based edge weights. For instance, 10% of the generated edge weights were set to two units, 1% was set to three units, 0.1% was set to four units, and so on. Such a weighted graph produced non-shortest paths during path tracing, which replicated observation of the path characteristics observed in [49]. Additionally, as not all routers are responsive, such unresponsive routers are marked as an asterisk ('*') in the traceroute output [15]. Hence, we consider when a subset of routers is set as unresponsive routers with a power-law length distribution based on the measurements from [52]. In particular, we generated path traces when approximately 10% had one, 1% had two consecutive, 0.1% had three consecutive, and so on unresponsive routers in the path. Then, we performed ASIAR on the sampled networks with every possible combination of the parameters to optimize the precision and recall measures of the subnetwork and IP alias resolution with respect to the ground truth. We also generated network topologies of 10,000 routers involving about 19,000 IP addresses and 6000 subnets in a similar way.

The IP distribution for the 1000 and 10,000 node networks are provided in Figure 5 using a Hilbert curve plot. Yellow marks are the IPs assigned, the red marks are the IP space left empty while increasing the mask of the subnet needed, and the black marks are the unassigned IPs.

*7.2. Parameter Space Exploration*

In ASIAR, parameters that need to be optimized for subnetwork detection are the Completeness threshold used for checking the completeness of subnets and the Pair-error and Group-error thresholds used to allow a certain amount of error in the number of distance matches for the reference points. These thresholds provide the flexibility to tune the system for performing subnetwork resolution with varying trace density. Additionally, different modes of IP alias resolution conditions could be combined arbitrarily for 72 different sanity checks. Hence, it is essential to make sure that suitable values are selected for these thresholds and optimal sanity checks are performed to have a system that produces accurate resolution results.

We tested every combination of the three parameters to optimize the subnetwork resolution thresholds. We explored the parameter space for different sized networks to find the best solutions that would allow ASIAR to perform well on various networks. However, as shown below, depending on the network sample characteristics, the utilized p arameters differed. Hence, we introduced an evolutionary computing approach in Section 6.2 to determine the parameter set based on the sampled network's topological characteristics.

We calculated the average of the precision and recall of every sample and selected the parameter combination that gave the highest results. The precision, recall, and F-measure of different parameters are provided in Table 2 for the completeness condition with the subnetwork accuracy condition, Table 3 for the completeness and subnetwork distance conditions, and Table 4 for a combination of all. Note that the equation for precision is TP/(TP + FP) and recall is TP/(TP + FN), where TP is the number of IP pairs correctly identified as a pair, FP is the number of IP pairs that are not considered to be a pair although they are, and FN is the number of IP pairs that are considered to be a pair although they are not. In each analysis, the completeness condition is essential to prevent reliance on subnetworks with sparse data. The tables display the optimal parameters discovered for each sample size after examining all combinations in 10% increments. Although a Genetic Algorithm (GA) is used for efficiency in finding the best solutions, a full search was initially conducted to gauge the impact of different condition thresholds. This comprehensive exploration helped to understand the contributions of various thresholds, ensuring the optimization is empirically grounded and aims to improve the accuracy of our methodology.

When considering the subnetwork accuracy condition, i.e., no two IPs of the subnetwork can appear in any trace unless they are consecutive; the only parameter that changes is the subnetwork completeness, as tabulated in Table 2. Varying the completeness threshold with 10% increments from 0% to 100%, we observe that when 100 IPs were traced, at least 10% to 30% of the IPs need to be detected in each subnetwork. When 1000 IPs were traced, at least 10% to 40% of the IPs need to be seen in each subnetwork, and when all the IPs were traced, at least 10% to 60% of the IPs need to be observed in each subnetwork depending on the number of sources. For networks with 1, 10, and 100 sources, the necessity to identify a minimum of 10% to 60% of IPs within each subnetwork varies according to the number of destinations. This requirement highlights that, as the network sample becomes denser, the threshold for what constitutes sufficient data completeness to ensure accuracy naturally rises. This observation underscores the critical relationship between the depth of network data collected and the confidence in the analysis performed.

**Table 2.** Subnetwork completeness condition parameter with the accuracy condition.

|  | Completeness | Precision | Recall | F-Measure |
|---|---|---|---|---|
| 1 × 100 | ≥10% | 54.5% | 84.4% | 65.2 % |
|  | ≥20% | 77.0% | 39.1% | 50.1% |
| 1 × 1000 | ≥40% | 82.8% | 86.4% | 84.4% |
| 1 × all | ≥60% | 75.9% | 66.2% | 70.6% |
| 10 × 100 | ≥20% | 81.0% | 82.6% | 81.4% |
| 10 × 1000 | ≥50% | 91.0% | 80.6% | 85.4% |
| 10 × all | ≥60% | 75.8% | 68.0% | 71.7% |
| 100 × 100 | ≥20% | 70.3% | 91.8% | 79.3% |
|  | ≥10% | 57.9% | 98.6% | 72.5% |
|  | ≥30% | 80.5% | 62.8% | 70.5% |
| 100 × 1000 | ≥50% | 91.6% | 87.5% | 89.4% |
| 100 × all | ≥50% | 71.7% | 89.1% | 79.4% |
|  | ≥60% | 85.5% | 67.9% | 75.7% |
| 1000 × 1000 | ≥10% | 98.4% | 100% | 99.2% |
| 1000 × all | ≥10% | 99.0% | 100% | 99.5% |

**Table 3.** Subnetwork completeness and distance condition parameters.

|  | Completeness | Pair Error | Group Error | Precision | Recall | F-Measure |
|---|---|---|---|---|---|---|
| 1 × 100 | ≥10 % | ≤90% | ≤90% | 56.1% | 84.4% | 66.2% |
|  | ≥10% | ≤90% | ≤100% | 55.3% | 84.4% | 65.6% |
|  | ≥10% | ≤100% | ≤100% | 52.8% | 84.4% | 64.0% |
|  | ≥20% | ≤90% | ≤90% | 78.1% | 39.1% | 50.3% |
| 1 × 1000 | ≥40% | ≤90% | ≤90% | 89.1% | 86.4% | 87.7% |
| 1 × all | ≥50% | ≤ 90% | ≤90% | 65.7% | 86.7% | 74.7% |
|  | ≥60% | ≤90% | ≤90% | 81.9% | 66.2% | 73.1% |
| 10 × 100 | ≥20% | 0% | 0% | 82.5% | 82.6% | 82.2% |
| 10 × 1000 | ≥40% | 0% | 0% | 91.0% | 92.5% | 91.6% |
| 10 × all | ≥50% | 0% | 0% | 85.1% | 89.1% | 87.0% |
| 100 × 100 | ≥20% | 0% | 0% | 72.3% | 91.8% | 80.6% |
|  | ≥10% | 0% | 0% | 61.3% | 98.7% | 75.1% |
|  | ≥0% | 0% | 0% | 58.7% | 100% | 73.3% |
|  | ≥30% | 0% | 0% | 81.8% | 62.8% | 71.0% |
| 100 × 1000 | ≥40% | 0% | 0% | 95.5% | 94.2% | 94.8% |
| 100 × all | ≥20% | 0% | 0% | 92.2% | 98.4% | 95.2% |
| 1000 × 1000 | ≥10% | 0% | 0% | 98.7% | 94.8% | 96.6% |
| 1000 × all | ≥10% | 0% | 0% | 99.5% | 86.8% | 92.3% |

**Table 4.** Subnetwork completeness and distance condition parameters with the accuracy condition.

|  | Completeness | Pair Error | Group Error | Precision | Recall | F-Measure |
|---|---|---|---|---|---|---|
| 1 × 100 | ≥10% | ≤90% | ≤90% | 56.1% | 84.4% | 66.2% |
|  | ≥10% | ≤100% | ≤100% | 54.5% | 84.4% | 65.2% |
|  | ≥20% | ≤90% | ≤90% | 78.1% | 39.1% | 50.3% |
| 1 × 1000 | ≥40% | ≤90% | ≤90% | 89.1% | 86.4% | 87.7% |
| 1 × all | ≥50% | ≤90% | ≤90% | 65.7% | 86.7% | 74.7% |
|  | ≥60% | ≤90% | ≤90% | 81.9% | 66.2% | 73.1% |
| 10 × 100 | ≥20% | 0% | 0% | 82.5% | 82.6% | 82.2% |
| 10 × 1000 | ≥40% | 0% | 0% | 91.0% | 92.5% | 91.6% |
| 10 × all | ≥50% | 0% | 0% | 85.1% | 89.1% | 87.0% |
| 100 × 100 | ≥20% | 0% | 0% | 71.8% | 91.8% | 80.3% |
| 100 × 1000 | ≥40% | 0% | 0% | 95.5% | 94.2% | 94.8% |
| 100 × all | ≥20% | 0% | 0% | 92.0% | 98.4% | 95.1% |
| 1000 × 1000 | ≥10% | ≤10% | ≤10% | 98.5% | 100% | 99.3% |
| 1000 × all | ≥10% | ≤10% | ≤10% | 99.5% | 100% | 99.7% |

When considering the subnetwork distance condition, we have two parameters for the pair error rate and group error rate, in addition to the subnetwork completeness. While the completeness threshold is a lower bound, error thresholds are upper bound thresholds. In Table 3, we observe similar completeness ranges as in Table 2, where the subnetwork accuracy condition was considered. In the distance error thresholds, we observe that in networks with a single vantage point, the error parameters allow for 90% to 100% of distances to be mismatched. This indicates they do not affect the results, since there only exists a single vantage point (VP). In networks with 10 or more VPs, we observe that there could be no error with the distances of subnet IPs with respect to the reference points. Overall, as the sampling density increases, the error parameters become more stringent.

When considering the subnetwork accuracy and distance conditions along with the completeness condition in Table 4, we observe similar parameters to that of Table 3 without the accuracy condition. The precision and recall for both cases are very similar, which indicates that the accuracy condition does not contribute much over the distance condition. The highest contribution of the accuracy condition is observed when the number of VPs is 1000.

Finally, Table 5 provides the order of modes used for different network samples. In order to determine the orders that yield the highest accuracy, we employ an iterative consideration of all possible modes. We initially test how well each of the modes performs by themselves and select the one that has the highest precision. If multiple modes were yielding the highest accuracy, we prioritize the more strict ones to assure high precision in IP alias resolution. However, it is important to keep appending modes to increase the recall rate. Once we determine the first mode to be considered, we append the modes one by one to the previous modes, and observe whether or not the precision decreases immensely. It is possible that the precision decreases slightly, while the recall increases considerably. In those cases, we continue appending best of the remaining modes. We iterate until we observe a large decrease in the precision and terminate the mode selection process. Table 5 lists the modes in the order of which yielded the highest precision and recall values for the synthetic networks we generated. In most samples, we observe a consistency in the order of modes selected. However, when the network contains a very small number of sources or destinations, different modes became more useful to obtain high resolution accuracy.

**Table 5.** IP alias resolution mode order for different samples.

| No. of src IPs | No. of dst IPs | Modes |
|---|---|---|
| 1 | 100 or 1000 | 1, 2, 3, 4, 5 |
| | all | 1, 6, 2, 7, 3, 8, 4, 9, 10 |
| 10 | 100 | 5 |
| | 1000 | 1, 2, 3 |
| | all | 1, 2, 3, 4 |
| 100 | 100 | 5 |
| | 1000 or all | 1, 2, 3, 4 |
| 1000 | 1000 or all | 1, 2, 3, 4 |

### 7.3. ASIAR Performance with Synthetic Network

In this section, we assessed the performance of ASIAR with the identified parameters from Section 7.2 on a new set of networks. We also analyzed performance with differing network sizes and performed IP alias resolution with the well-known Kapar tool.

Figure 6 presents the ASIAR subnetwork and IP alias resolution performance results along with the Kapar IP alias resolution for five networks of 1000 nodes. Each subfigure presents the precision alongside the recall of networks with a given number of sources and destinations. In the figures, the x-axis presents the precision and recall values. Furthermore, the y-axis indicates the synthetic network characteristic, where 1 is for the unit network, 1* is for the unit network with unresponsive routers, w is for the weighted network, and w* is for the weighted network with unresponsive routers. Each box plot shows the five-number summary statistics of (min, first quartile, median, third quartile, and max), where a narrower band indicates a more consistent result. Note that while there are 1000 router nodes, they contain approximately 1900 interfaces, each with a unique IP address. We generated network samples with unit edge weights where all path traces are the shortest paths and weighted edges where some of the paths are not the shortest. We also considered when about 12% of routers were set as unresponsive, i.e., their IP addresses were hidden, based on the measurements of [53].

With the subnetwork resolution, we observe that as the number of IPs traced increases, the precision decreases if the number of VPs does not increase at the same time. We also observe that as the number of VPs increases, the precision increases. We observe a minimum of 84% precision and 64% recall when the number of VPs is small. However, a minimum of 91% precision and 86% recall is reached when the number of VPs is increased. As traces get denser where both the number of VPs and destinations are increased, ASIAR was able to perform subnetwork resolution at a minimum of 95% precision and recall. We achieve 100% recall when all of the IPs are traced.

In IP alias resolution, we observe that ASIAR increases the precision as the number of VPs increase, and it increases the recall with an increasing number of destinations. Even with 10 VPs, we achieve a minimum of 87% precision. Although Kapar achieves higher recall in many cases, it obtains very low precision in almost all of those cases. In all modes, ASIAR prioritizes the IP alias resolution precision while obtaining an acceptable level of recall. As the network gets denser, ASIAR reaches a minimum of 96% precision when half of the IPs are traced toward.

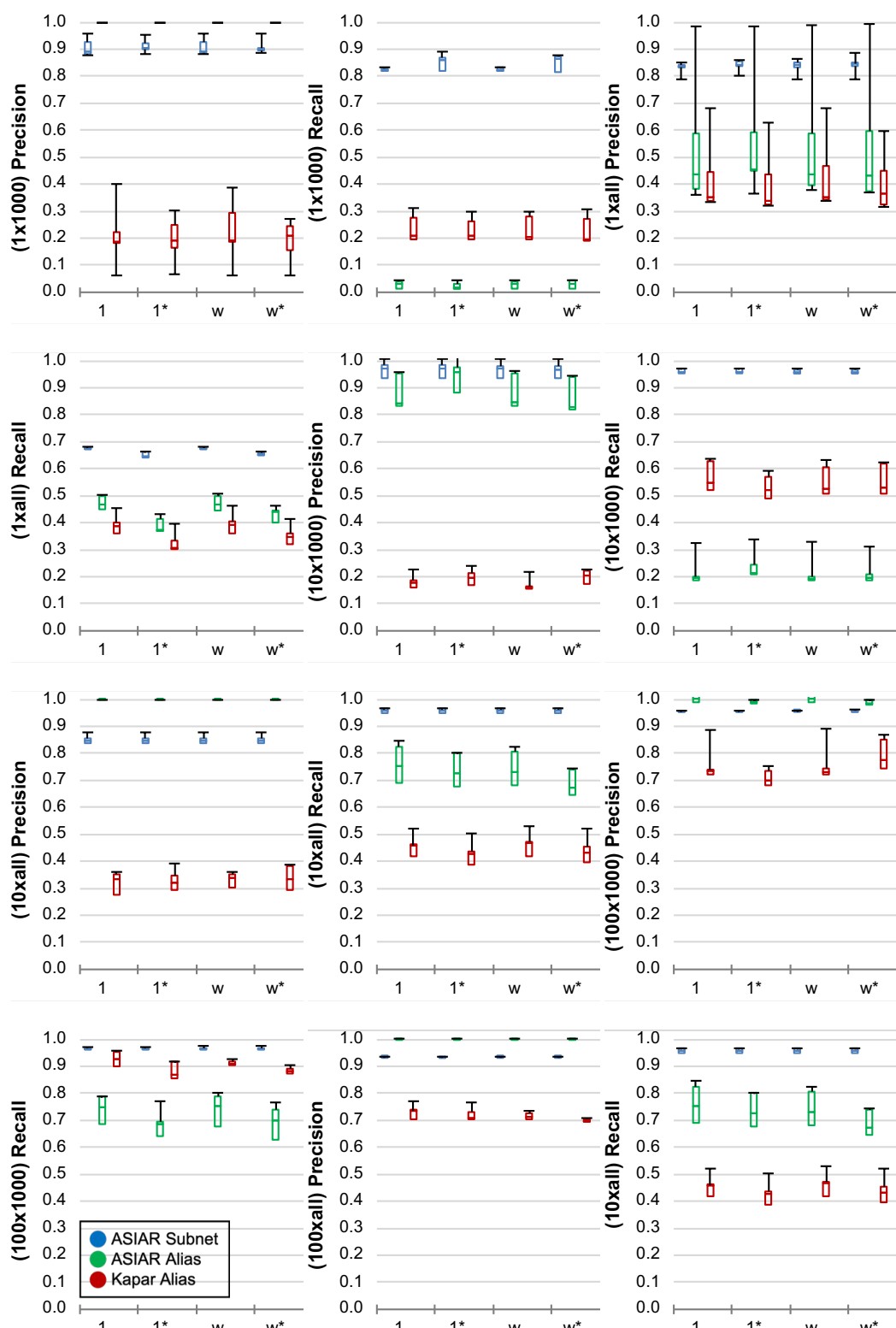

**Figure 6.** Simulation with varying sources and destinations on 1000-node networks.

After analyzing performance of ASIAR on newly generated networks based on parameters selected from the prior analysis, we tested ASIAR and Kapar on a larger network where we did not perform parameter selection. We generated a network of 10,000 nodes and tested both tools on this network. Since it was computationally complicated to obtain 10,000-node networks and sample traces, we tested ASIAR and Kapar on a single network.

During our discussion, we will focus more on the results obtained from the weighted unresponsive traces, since those are the most difficult ones to resolve.

Figures 7 and 8 present the results for 1* unit and w* weighted networks, respectively, where unresponsive routers are considered. In the figures, the x-axis indicate the obtained precision or recall. Similar to what we observe with 1000-node networks, ASIAR is able to keep its precision high in all cases. We also observe that, as the traces become denser, Kapar increases its recall while sacrificing the precision. ASIAR is able to keep precision as high as possible in almost all cases while having an acceptable level of recall even without parameter optimization.

Our analysis indicates that ASIAR effectively adapts to diverse network characteristics, carefully choosing parameters that guarantee high precision in all analyzed networks. Conversely, Kapar, although it exhibits higher recall in networks with lower density, struggles with precision, underlining ASIAR's enhanced adaptability and superior accuracy for thorough network analysis tasks.

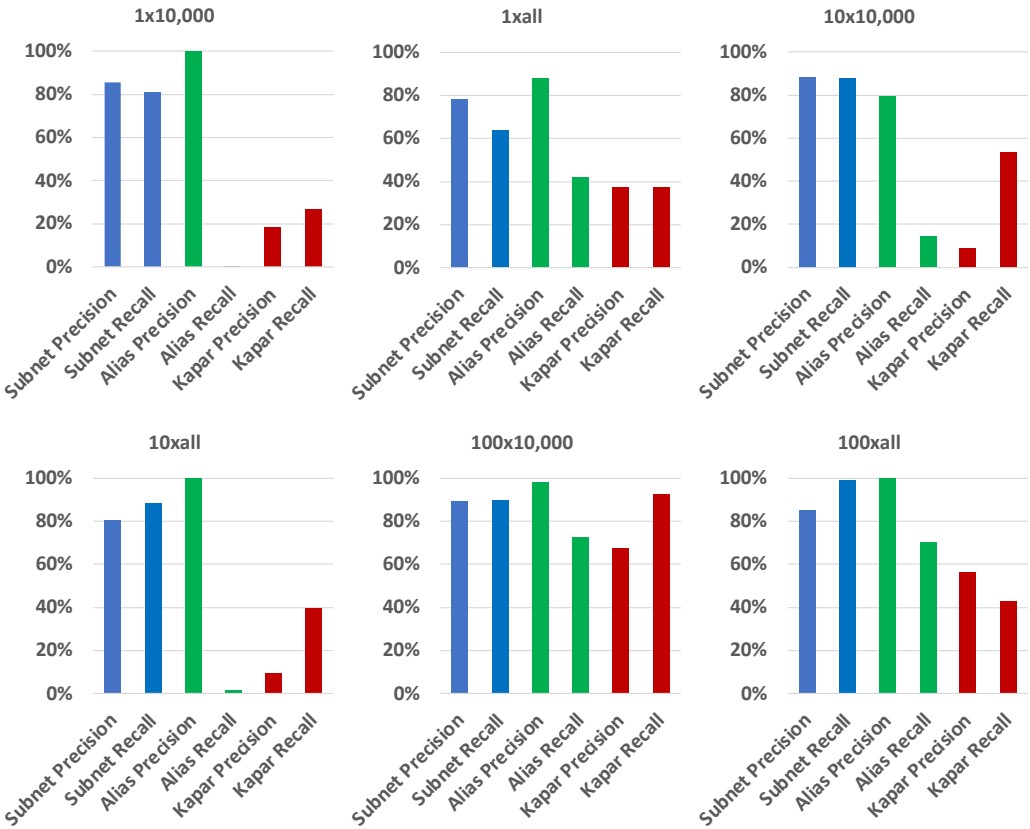

**Figure 7.** Simulation of a 10,000-node network with 1* unit weights and unresponsive routers.

### 7.4. Evaluation on a Genuine Network

In this section, we present results for a real network with multiple points of presence in Nevada for which we obtained detailed subnetwork and IP alias information. Note that while system administrators have network maps, they typically do not contain detailed information on subnetworks and router interfaces. Even if such detailed maps are available, they are not made public or shared with researchers. We obtained traces traversing the network from the Internet Measurement (IM) platform [13] we had developed, without any tweaks to the system for the destination AS. IM collects publicly available topology measurement datasets and BGP announcements, and traces toward each of the IP address of an AS observed in these datasets using 20 to 30 vantage points deployed globally. We simply filtered trace segments belonging to the destination AS for the analysis.

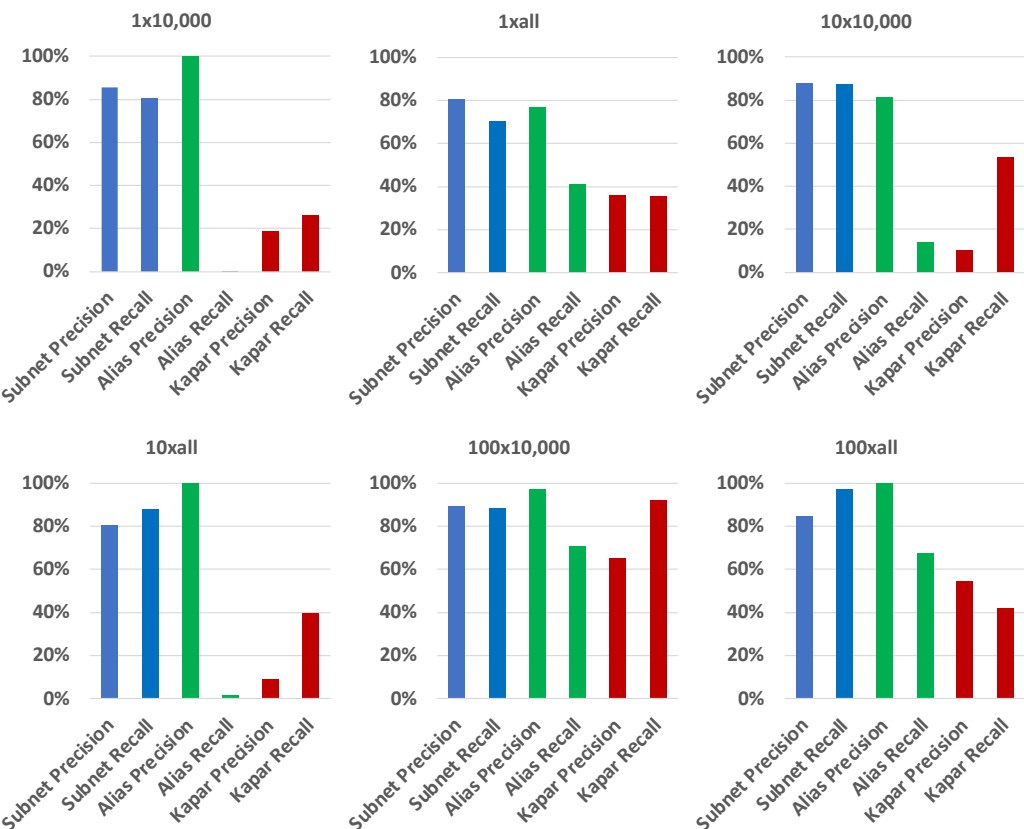

**Figure 8.** Simulation of a 10,000-node network with w* weighted edges and unresponsive routers.

Our methodology began with deploying MIDAR on the network traces to establish an initial benchmark against the network's verified ground truth. MIDAR achieved an impressive precision rate of 99.5%, although its recall rate stood at 42.5%, indicating a high level of accuracy in identifying true aliases, but also suggesting room for improvement in uncovering a more comprehensive set of alias pairs within the network. To further refine our understanding and assessment of network analysis tools, we extended our examination to include ASIAR and Kapar. This phase involved conducting performance tests on these tools within the same network environment, implementing scenarios where they either had access to the alias pairs previously identified by MIDAR or operated without this pre-identified data. This dual approach allowed us to to explore the synergistic potential that might emerge from leveraging MIDAR's findings as a foundational dataset. Through this layered testing strategy, we aimed to comprehensively evaluate the effectiveness of each tool in enhancing the precision and recall rates of network alias detection, thereby contributing valuable insights into their operational efficacy and potential integrations for advanced network topology analysis.

As seen in Table 6, ASIAR outperforms Kapar in both cases. Similar to the synthetic networks, ASIAR prioritizes the precision over the recall, while Kapar yields higher recall than the precision. Additionally, Kapar benefits significantly from the input from MIDAR with considerable improvement in both the precision and recall. On the other hand, ASIAR has mixed results with MIDAR input, where recall improves at the expense of precision. Overall, ASIAR achieves much higher precision and recall than Kapar.

**Table 6.** Subnetwork and IP alias resolution on a genuine network.

| | Without MIDAR | | With MIDAR | |
|---|---|---|---|---|
| | **Precision** | **Recall** | **Precision** | **Recall** |
| ASIAR Subnetwork | 87.0% | 78.0% | 86.0% | 78.0% |
| ASIAR IP Alias | 88.0% | 71.4% | 82.3% | 78.0% |
| Kapar IP Alias | 48.1% | 53.5% | 65.5% | 75.5% |

## 8. Conclusions

In this paper, we presented an efficient implementation of the Analytical Subnetwork and IP Alias Resolution (ASIAR) technique, significantly advancing the field by introducing additional sanity checks aimed at improving resolution accuracy. Through a detailed analysis of these sanity checks across varied network samples, we showcased their efficacy in enhancing IP alias resolution. Furthermore, we unveiled a comprehensive analytical tool that employs evolutionary computing to fine-tune ASIAR's parameters based on analogous synthetic networks. Our findings demonstrate that ASIAR outperforms the current state-of-the-art analytical IP alias resolution tools in both synthetic and genuine networks. ASIAR maintains superior precision and recall rates under complex conditions, including dense trace data, non-shortest path traces, and encounters with non-responsive routers.

ASIAR's adaptability extends to its ability to dynamically adjust the stringency of sanity checks based on the specific characteristics of the sampled measurement data, making it a versatile solution for various network samples. This flexibility contrasts with previous approaches that applied a uniform toolset across all networks, overlooking the nuanced differences in topology samples. By tailoring sanity checks and parameters optimization to synthetic networks with comparable characteristics, ASIAR offers a more nuanced and effective approach to analytical alias resolution, emphasizing the importance of customizing methods to suit the intricacies of each network scenario.

In our analysis of synthetic network data, we conducted extensive trace experiments to evaluate the impact of network sampling on subnetwork and IP alias precision and recall. To enhance accuracy and flexibility, we introduced three parameters within the ASIAR tool, allowing us to navigate a nuanced parameter space and identify optimal threshold values for our system. Through this exploration, we consistently achieved impressive precision and recall rates for both subnetwork and IP alias resolution tasks. Notably, when compared to Kapar, ASIAR demonstrated a remarkable ability to maximize recall without compromising precision. Particularly in denser trace scenarios, ASIAR exhibited significantly higher precision and recall rates, reaching 99% and 70% respectively, compared to Kapar's 56% precision and 43% recall. This highlights ASIAR's effectiveness in handling complex network data with superior performance metrics.

**Funding:** This research received no external funding.

**Institutional Review Board Statement:** Not applicable.

**Informed Consent Statement:** Not applicable.

**Data Availability Statement:** Data are contained within the article.

**Acknowledgments:** I extend my heartfelt gratitude to Mehmet Gunes for their invaluable assistance and guidance throughout the course of this research. Their expertise and insightful perspectives have been crucial in the development of this paper. Gunes's encouragement and constructive feedback significantly contributed to refining my work, and their dedication to academic excellence has been a constant source of inspiration. This project benefited immensely from their profound knowledge and mentorship, for which I am deeply grateful.

**Conflicts of Interest:** The authors declare no conflicts of interest.

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
