# Peer review of "Analytical Subnetwork and IP Alias Resolution for Network Tomography Using Path Traces"

_electronics, doi:10.3390/electronics13081426_

Round 1

Reviewer 1 Report

Comments and Suggestions for Authors

Review Report on the manuscript with the title "ASIAR: Analytical Subnetwork and IP Alias Resolution for Network T omography using Path T races".

 In this paper, the authors have presented a comprehensive Analytical Subnetwork and Alias IP Resolution (ASIAR) method that relies on the analytical analysis of path traces to infer underlying subnetworks and router interfaces,and explore how different network sampling issues affect the analytical resolution, and analyze the accuracy of the ASIAR on synthetic and genuine networks. The explanation process is clear and provides experimental data and result analysis, but it needs to be modified in the content and layout structure of the article.

The authors should introduce some recent work on networks structural properties and networks algorithms, such as the titled "Analyses of some structural properties on a class of hierarchical scale-free networks. Fractals " and "Mean First-Passage Time and Robustness of Complex Cellular Mobile Communication Network."

Page 1: In Introduction, it is necessary to highlight the specific value of references to this article, and the details can be concise without too much description.

Page 2: It should be noted that to reduce colloquial expressions, sentences like "Our results indicate that..." need to be revised.

Page 6:For the Figure 3, it may be necessary to make adjustments in color matching and overall structural layout to make the picture more beautiful.

Page 10 and Page 11:The structural distribution of "Analytic IP Alias Resolution" content and the layout of algorithm modules need to be adjusted to make the whole part more compact.

Page 22:In the section "Conclusion", it is a summary of the whole article, which needs to be as comprehensive and concise as possible, focusing on the conclusions drawn from the analysis.The first paragraph "Network topology mapping is the process of..." in this part needs to be considered whether it can be expressed concisely.

Hence, I recommend this paper for publication after a major revision.

Comments on the Quality of English Language

Review Report on the manuscript with the title "ASIAR: Analytical Subnetwork and IP Alias Resolution for Network T omography using Path T races".

 In this paper, the authors have presented a comprehensive Analytical Subnetwork and Alias IP Resolution (ASIAR) method that relies on the analytical analysis of path traces to infer underlying subnetworks and router interfaces,and explore how different network sampling issues affect the analytical resolution, and analyze the accuracy of the ASIAR on synthetic and genuine networks. The explanation process is clear and provides experimental data and result analysis, but it needs to be modified in the content and layout structure of the article.

The authors should introduce some recent work on networks structural properties and networks algorithms, such as the titled "Analyses of some structural properties on a class of hierarchical scale-free networks. Fractals " and "Mean First-Passage Time and Robustness of Complex Cellular Mobile Communication Network."

Page 1: In Introduction, it is necessary to highlight the specific value of references to this article, and the details can be concise without too much description.

Page 2: It should be noted that to reduce colloquial expressions, sentences like "Our results indicate that..." need to be revised.

Page 6:For the Figure 3, it may be necessary to make adjustments in color matching and overall structural layout to make the picture more beautiful.

Page 10 and Page 11:The structural distribution of "Analytic IP Alias Resolution" content and the layout of algorithm modules need to be adjusted to make the whole part more compact.

Page 22:In the section "Conclusion", it is a summary of the whole article, which needs to be as comprehensive and concise as possible, focusing on the conclusions drawn from the analysis.The first paragraph "Network topology mapping is the process of..." in this part needs to be considered whether it can be expressed concisely.

Hence, I recommend this paper for publication after a major revision.

Author Response

Comment:

The authors should introduce some recent work on networks structural properties and networks algorithms, such as the titled "Analyses of some structural properties on a class of hierarchical scale-free networks. Fractals " and "Mean First-Passage Time and Robustness of Complex Cellular Mobile Communication Network."

Answer:

Thank you for your constructive comments and for suggesting the inclusion of recent work on network structural properties and algorithms, specifically referencing "Analyses of Some Structural Properties on a Class of Hierarchical Scale-free Networks" and "Mean First-Passage Time and Robustness of Complex Cellular Mobile Communication Network." After carefully reviewing these studies, I found that while both papers offer significant insights into the field of network science, their direct relevance to my work on the Analytical Subnetwork and Alias IP Resolution (ASIAR) method varies due to the distinct focus of each study. The first paper presents an in-depth examination of hierarchical networks with scale-free and fractal structures, emphasizing mathematical and topological analysis. Although this work contributes valuable knowledge on the structural properties of networks, its application to the specific challenges addressed by the ASIAR method—namely, the analytical resolution of IP aliases and mapping of network topologies—is limited. My method is more closely aligned with network engineering tasks that require an understanding of the practical aspects of network topology discovery. Similarly, the second suggested study investigates the dynamics and robustness of cellular mobile communication networks using a fractal six-star network model. This research provides important insights into the performance and resilience of cellular networks but addresses a different aspect of network science than my focus on topology mapping and IP alias resolution. Nevertheless, I acknowledge the importance of situating my work within the broader context of recent advancements in network science. Therefore, in addition to considering the suggested papers, I have incorporated references to a few more recent studies that are more closely aligned with the challenges and objectives of the ASIAR method. I appreciate your guidance in strengthening the context and relevance of my research and hope that my response and the adjustments made to my manuscript adequately address your concerns.

Comment:

It should be noted that to reduce colloquial expressions, sentences like "Our results indicate that..." need to be revised.

Answer:

Thank you for your insightful comment regarding the use of colloquial expressions in my manuscript. I have carefully reviewed the text and revised sentences such as "Our results indicate that..." to ensure that the language in the paper adheres more closely to the formal tone expected in academic writing. These modifications have been made throughout the document to enhance its clarity and professionalism.

Comment:

For the Figure 3, it may be necessary to make adjustments in color matching and overall structural layout to make the picture more beautiful.

Answer:

Thank you for your feedback. In response to your suggestions, I have made a revision to the presentation of the figure. Specifically, I have updated the graphical representation of the routers within the network diagrams, ensuring that they are consistent across both graphs. This change not only enhances the visual harmony and coherence between the figures but also maintains the integrity and clarity of the information presented.

Comment:

The structural distribution of "Analytic IP Alias Resolution" content and the layout of algorithm modules need to be adjusted to make the whole part more compact.

Answer:

Thank you for your feedback. Taking your advice into consideration, I have carefully revised this section to ensure a more compact and cohesive presentation. I have integrated the algorithms directly into the text immediately following their first mention, which should provide a smoother reading experience and a clearer understanding of their application within the context of the discussion. This change avoids any interruption in the flow of the manuscript and allows the reader to see the practical implementation of the algorithms as they are introduced. I trust that these modifications address your concerns by improving the logical progression and compactness of the content, thereby enhancing the overall quality and readability of the paper.

Comment:

In the section "Conclusion", it is a summary of the whole article, which needs to be as comprehensive and concise as possible, focusing on the conclusions drawn from the analysis.The first paragraph "Network topology mapping is the process of..." in this part needs to be considered whether it can be expressed concisely.

Answer:

Thank you for your observations. Upon reflection, I concur with your assessment that the opening paragraph should succinctly encapsulate the essence of our findings rather than delve into an explanatory narrative. In response, I have removed the detailed exposition on network topology mapping from the first paragraph. Instead, I have crafted a new paragraph that directly summarizes the pivotal conclusions derived from our analysis. This revised approach aligns with the aim to provide a comprehensive yet concise summation of our work. Additionally, to offer further value to our readers and the research community, I have included a "Future Work" section. This new section outlines potential avenues for subsequent research, building upon the foundation laid by the findings presented in this paper.

Reviewer 2 Report

Comments and Suggestions for Authors

Authors in the paper present an additional technique for sanity checks to dynamically adjust stringiness for analytical alias resolution.

The authors provide the github link that is so valuable. But it can be more useful if they provide a Readme and add comment of each part of their code. 

The are many new research and contributes in the state of art. The author can review those works too. the most recent paper is from 2020.

Comments on the Quality of English Language

it is acceptable. 

Author Response

Comment 1: The authors provide the github link that is so valuable. But it can be more useful if they provide a Readme and add comment of each part of their code. 

Response: Thank you for your constructive feedback regarding the accessibility and usability of our code repository. Taking your suggestion into account, I have addressed this by creating a comprehensive README file for the GitHub repository. This README includes a detailed overview of the repository's structure, instructions for installing necessary dependencies, running the code, and explanations of the main components and functionalities of our project. I believe these enhancements significantly improve the usability of our repository and facilitate a deeper understanding of the methodologies employed in our study.

Comment 2: The are many new research and contributes in the state of art. The author can review those works too. the most recent paper is from 2020.

Response: Thank you for your valuable feedback. Concerning the age of references, I have incorporated the most current studies available. However, due to the novel nature of this research field, there is a scarcity of recent literature. This scarcity has necessitated reliance on foundational works and seminal papers, some of which may be older, to establish the context and significance of the current study. Nonetheless, I have endeavored to update the literature review with the latest relevant studies to reflect ongoing developments and insights in the field, emphasizing its novelty and the contribution of this work to advancing knowledge.

Reviewer 3 Report

Comments and Suggestions for Authors

1. Figures are not located immediately after they are mentioned in the text. They are arranged randomly. They should be located immediately after being mentioned in the text;

2. The proposed scripts for algorithms 1 to 4 are also placed randomly. Each script must be placed in the correct place in the text. For example, line 328 mentions Algorithm 1 - "Algorithm 1 presents the pseudocode for the identification of subnetworks." The next line should include the proposed script for Algorithm 1 and then start with the description of the individual lines of the script;

3. Tables should also be placed immediately after they are mentioned in the text;

4. The literature is not arranged sequentially. Also, the references are very old, some from 15 years ago. All references must be replaced with new, at least from the last 4 years.

Author Response

Comment 1: Figures are not located immediately after they are mentioned in the text. They are arranged randomly. They should be located immediately after being mentioned in the text.

Response: Thank you for bringing this to my attention. I have made a concerted effort to place figures immediately after they are mentioned in the text to enhance readability and coherence. However, due to the constraints imposed by LaTeX formatting and the dynamic nature of document compilation, achieving the exact placement can sometimes be challenging. Despite these limitations, I have revised the document to improve the positioning of figures as closely as possible to their initial mention, ensuring a smoother flow of information.

Comment 2: The proposed scripts for algorithms 1 to 4 are also placed randomly. Each script must be placed in the correct place in the text. For example, line 328 mentions Algorithm 1 - "Algorithm 1 presents the pseudocode for the identification of subnetworks." The next line should include the proposed script for Algorithm 1 and then start with the description of the individual lines of the script.

Response: I appreciate your feedback on the organization of the algorithm scripts within the manuscript. Following your suggestion, I have meticulously reviewed and repositioned each script to directly follow its corresponding mention in the text, especially for the identified instances such as line 328 concerning Algorithm 1. This reorganization aims to facilitate a better understanding of each algorithm's structure and functionality, directly linking the discussion in the text with the respective pseudocode.

Comment 3: Tables should also be placed immediately after they are mentioned in the text.

Response: Thank you for pointing out the placement of tables within the manuscript. Similar to the issue with figures, I aimed to place tables immediately after their mention to maintain textual coherence and aid in the reader's comprehension. However, the automatic layout adjustments of LaTeX sometimes prevent exact placements. In response to your comment, I have adjusted the document to improve the proximity of tables to their references in the text, ensuring they are as close as feasibly possible given formatting constraints.

Comment 4: The literature is not arranged sequentially. Also, the references are very old, some from 15 years ago. All references must be replaced with new, at least from the last 4 years.

Response: I understand your concerns regarding the sequencing and recency of the literature cited in my manuscript. The arrangement of references is automatically managed by LaTeX, which organizes them based on the citation style and sequence of citations within the text. Concerning the age of references, I have incorporated the most current studies available. However, due to the novel nature of this research field, there is a scarcity of recent literature. This scarcity has necessitated reliance on foundational works and seminal papers, some of which may be older, to establish the context and significance of the current study. Nonetheless, I have endeavored to update the literature review with the latest relevant studies to reflect ongoing developments and insights in the field, emphasizing its novelty and the contribution of this work to advancing knowledge.

Round 2

Reviewer 2 Report

Comments and Suggestions for Authors

The README file is not a complete description of all files in github. Please remove extra files from github or complete the readme file.

Comments on the Quality of English Language

github is not applicable.

Author Response

Thank you very much for your insightful feedback and for taking the time to review my GitHub repository. I understand the importance of maintaining clear and comprehensive documentation, and I appreciate your guidance in identifying this oversight.

In response to your comment, I have updated the README file to include detailed descriptions of all the library files currently present in the repository. Additionally, I reviewed the repository to ensure that all files included are essential to the project. This step was taken to adhere to your suggestion of removing any extraneous files that might have been present.

Thank you once again for your contribution to refining my project.

Reviewer 3 Report

Comments and Suggestions for Authors

I have only a few minor remarks:

1. Subsection 3.3 must begin with some text. Therefore, it is good that figure 3 should be moved immediately after "Varying the resolution success rate, we showed that the IP alias resolution process has a significant impact on the observed topological characteristics [19]"

2. "Algorithm 2 checks if IPs within a candidate subnet...................." is interrupted by the picture of algorithm 3 and then the text continues. Correct it.

3. Move figure 4 immediately after the text where figure 4 is mentioned.

4. Tables 2 and 3 to be moved immediately after the text where they are mentioned.

5. Figure 8 should be moved before the beginning of section 8, immediately after table 6. There should be no tables or figures in the "Conclusion" section.

It is suggested to the author not to use the LaTex in the future, as the platform itself organizes the arrangement of the figures and tables. This will save the author additional editing.

Best regards.

Author Response

Dear Reviewer,

Thank you for your valuable feedback and suggestions regarding the organization of figures, tables, and algorithms in my manuscript.

In response to your recommendations, I have carefully revised the manuscript to ensure that all figures, tables, and textual elements are correctly placed for optimal reader comprehension. I did my utmost during the revision process. I paid close attention to ensure that sentences were not inadvertently cut off or obscured by the placement of figures and tables by updating the text when necessary.

Regarding your suggestion about the use of LaTeX, I understand the concerns raised and will consider your advice for future submissions.

Thank you again for your insightful feedback and for aiding in the enhancement of my manuscript. I am hopeful that these revisions meet your approval and significantly improve the readability and quality of the work.

Best regards.